# Study protocol: examining the impacts of COVID-19 mitigation measures on pregnancy and birth outcomes in Scotland—a linked administrative data study

Moritz Oberndorfer [1,2] Paul M Henery,[3] Ruth Dundas,[1] Alastair H Leyland,[1] Shantini Paranjothy [4] Sarah Jane Stock [3,5] Rachael Wood [3,5] Scott M Nelson,[6] Rachel Kearns,[6] Anna Pearce[1]

For numbered affiliations see end of article.

**Correspondence to**
Moritz Oberndorfer;
moritz.oberndorfer@muv.ac.at

## ABSTRACT

**Introduction** This protocol outlines aims to test the wider impacts of the COVID-19 pandemic on pregnancy and birth outcomes and inequalities in Scotland.

**Method and analysis** We will analyse Scottish linked administrative data for pregnancies and births before (March 2010 to March 2020) and during (April 2020 to October 2020) the pandemic. The Community Health Index database will be used to link the National Records of Scotland Births and the Scottish Morbidity Record 02. The data will include about 500 000 mother–child pairs. We will investigate population-level changes in maternal behaviour (smoking at antenatal care booking, infant feeding on discharge), pregnancy and birth outcomes (birth weight, preterm birth, Apgar score, stillbirth, neonatal death, pre-eclampsia) and service use (mode of delivery, mode of anaesthesia, neonatal unit admission) during the COVID-19 pandemic using two analytical approaches. First, we will estimate interrupted times series regression models to describe changes in outcomes comparing prepandemic with pandemic periods. Second, we will analyse the effect of COVID-19 mitigation measures on our outcomes in more detail by creating cumulative exposure variables for each mother–child pair using the Oxford COVID-19 Government Response Tracker. Thus, estimating a potential dose–response relationship between exposure to mitigation measures and our outcomes of interest as well as assessing if timing of exposure during pregnancy matters. Finally, we will assess inequalities in the effect of cumulative exposure to lockdown measures on outcomes using several axes of inequality: ethnicity/mother's country of birth, area deprivation (Scottish Index of Multiple Deprivation), urban-rural classification of residence, number of previous children, maternal social position (National Statistics Socioeconomic Classification) and parental relationship status.

**Ethics and dissemination** NHS Scotland Public Benefit and Privacy Panel for Health and Social Care scrutinised and approved the use of these data (1920-0097). Results of this study will be disseminated to the research community, practitioners, policy makers and the wider public.

## STRENGTHS AND LIMITATIONS OF THIS STUDY

⇒ We will use population-wide administrative data covering all mother–child pairs for children born in Scotland between March 2010 and October 2020 to study how population-level pregnancy and birth outcomes changed during the COVID-19 pandemic.

⇒ Using the Stringency Index recorded by the Oxford COVID-19 Government Response Tracker, we are able to calculate an individual level of cumulative exposure to pandemic mitigation measures for each mother–child pair in our data.

⇒ Our effect estimates will be biased if unmeasured factors changed routine data collection (patterns of missing or misclassified data), or—for postnatal outcomes—if the characteristics of live births during the COVID-19 pandemic had changed in a way that is associated with our outcomes of interest.

## INTRODUCTION

Early on in the COVID-19 pandemic, concerns were raised about the widespread and unequal impacts of social mitigation measures on health and the social determinants of health[1] including for children and families.[2 3] In this protocol, we focus on parents and children during pregnancy and at birth. Figure 1 outlines three key, interlinked mechanisms through which the wider pandemic (distinct from the risks of contracting the virus) may have had negative (and sometime positive) effects on this group. The first surrounds changes to health services. Pregnant women were identified as being particularly vulnerable to the severe effects of COVID-19, prompting early advice from the National Health Service (NHS) to adopt social distancing. This, alongside the strain put on health services by the wider pandemic, meant that the services and

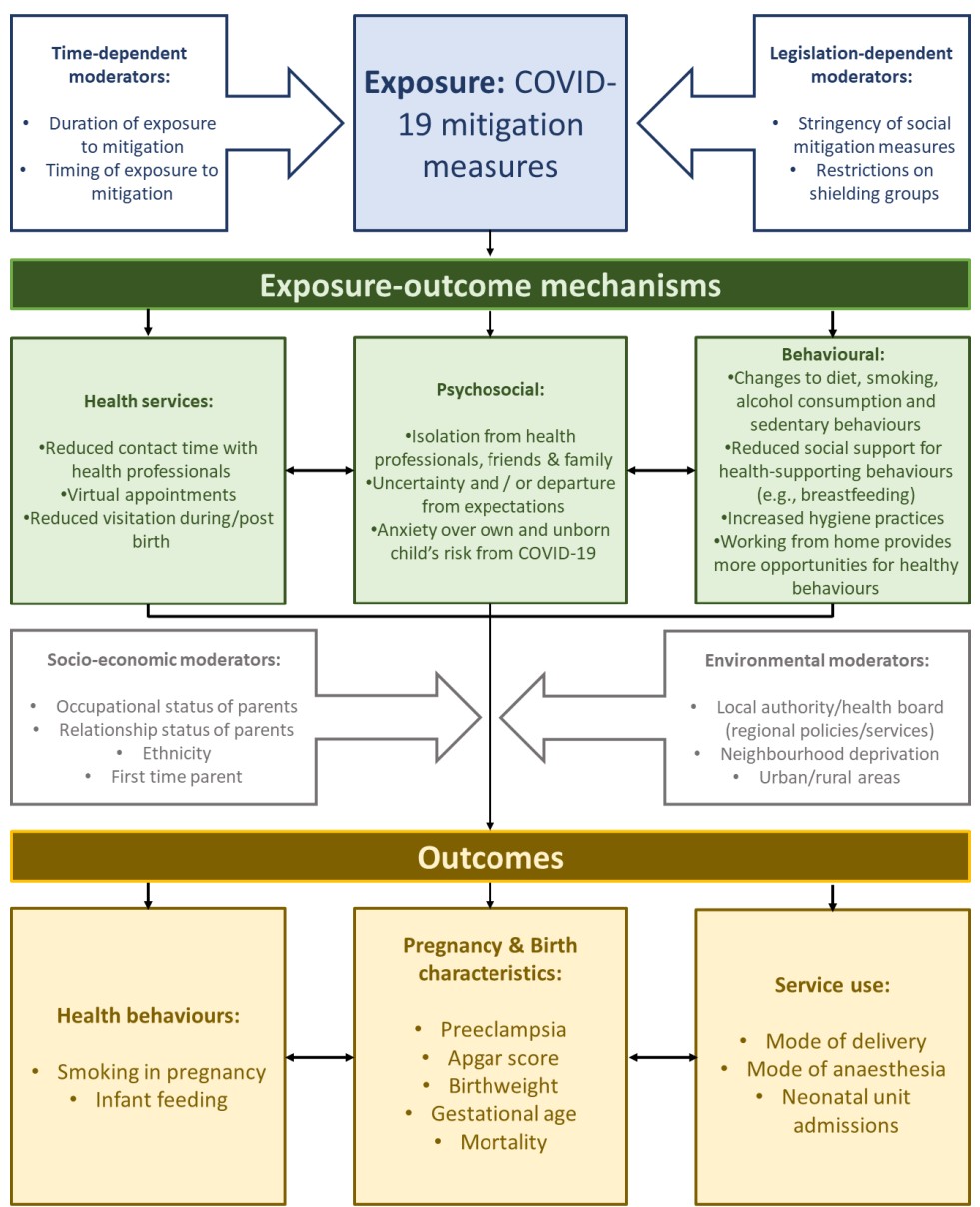

**Figure 1** Logic model demonstrating the mechanisms and moderators of the wider impacts of the pandemic on pregnancy and birth outcomes.

support for pregnant and new mothers dramatically changed.[4] Non-urgent procedures and contacts were cancelled, and resources diverted from elective to critical care. Guidance and services were quickly innovated to support new families, including the use of virtual technologies to provide health appointments, antenatal classes and hospital tours; mothers were supported to self-monitor glucose, urine and blood pressure at home; the provision of clinics in community settings increased. Partners were allowed in hospital only for the last stages of labour and no other visitors were permitted at any point during the hospital stay.[4] Although many of these restrictions have since eased, the services that young families receive have not fully returned to normal and uncertainty remains.

The second mechanism refers to psychosocial factors. Negative impacts of lockdown on mental well-being have

been documented, alongside increases in job loss, job insecurity and universal credit claims among the adult population.[5–9] Profound changes to services and birthing plans, the disruption of feeding intentions and expectations around parenthood, and anxiety around catching the virus, have led to increased uncertainty and feelings of isolation among pregnant mothers and new families causing psychological distress.[10 11]

Third, in the general population many health behaviours were affected, with diets becoming less healthy both in terms of quality and quantity[12] and alcohol consumption increasing, particularly among groups who were already high consumers.[13] Conversely, smoking has declined[5] and it has been hypothesised that working from home, lower exposure to air pollutants and better hygiene habits may have benefited fetal development and health.[14] Hospital support for breast feeding immediately after birth has

remained,[10] and breastfeeding rates on discharge have not necessarily been affected.[15] However, lack of support from friends and family, mother and baby groups and health professionals has been highlighted as a barrier to feeding after returning home.[10]

Our aim is not to test these different mechanisms, but to first establish the overall impacts of the pandemic on various mother and infant outcomes, and inequalities in these outcomes, in Scotland. This will provide a better understanding of potential future health challenges and to inform responses to the ongoing and any future pandemics. A comprehensive investigation of pregnancy and birth outcomes in Scotland during March to May 2020 (compared with 2 years previous) found that some procedural outcomes showed changes in the expected direction (eg, length of hospital stay decreased), but few changes in maternal and infant health outcomes.[15] Few signs of negative impacts (in high-income countries) have also been detected in international systematic reviews and meta-analyses,[14 16] with the exception of maternal mental health.[16] However, while the overall picture is positive, it remains plausible that these studies have overlooked differential effects occurring at the subgroup level. In the case of the three of proposed mechanisms discussed above, it is likely that some groups, including those from less advantaged social circumstances, first-time mothers and ethnic minority groups, have fared worse than others.[1 17] There are also some indications in the limited evidence base that birth and pregnancy have worsened from some groups and not others. For example, there was no change in stillbirths in England overall, but rates had increased in North England.[18] In the USA, newborn readmission rates among first-time mothers were higher after the pandemic, while multiparous women were less likely to experience preterm birth rates, low Apgar scores and hospital readmissions.[19] Furthermore, it is possible that early studies considering outcomes only at the very start of the pandemic may have overlooked impacts on expectant mothers who were exposed to social mitigation measures for longer durations of pregnancy.

We aim to estimate the wider impacts of the COVID-19 pandemic on pregnancy and birth outcomes and inequalities in Scotland. More specifically, we aim to estimate changes in health and pregnancy outcomes as a result of the pandemic. We will take a natural experiment approach to identify any step change trends in outcomes at the start of the pandemic, limiting our analyses to pregnancies which were conceived before the pandemic, to avoid introducing bias due to the changing sociodemographic characteristics of conceptions which occurred after the start of the pandemic.[20 21] As part of this aim, we will investigate whether exposure to mitigation measures had a differential effect on our outcomes across several axes of inequalities. Second, we aim to consider the cumulative effects of social mitigation measures across pregnancy. To this end, we will use the Stringency Index (SI; which measures the strictness of policies that primarily restrict people's behaviour) and compare cohorts with different lengths or intensity of exposure. Additionally, we aim to consider timing of exposure, as it is possible that, for some outcomes, any impacts of the stresses related to the pandemic and social mitigation measures might be greater during some trimesters of pregnancy than others.[22]

## METHODS

### Patient and public involvement

This secondary analysis of data will not directly involve the public or patients. Findings will be disseminated to relevant health professionals and interest groups to maximise benefits for service provision throughout Scotland.

### Study design and population

Our study population includes live births born between March 2010 and October 2020. More precisely, our population of interest consists of live births conceived before the pandemic who have not been exposed to COVID-19 mitigation measures in utero (live births between March 2010 and February 2020) and those who were conceived before the pandemic but were exposed to mitigation measures in utero (live births between March 2020 and October 2020).

We will employ two analytical approaches, each informed by the logic model in figure 1. In our first analytical approach, we will provide, using interrupted time series (ITS) regression models, a descriptive visualisation of how outcome variables changed between prepandemic (March 2010 to February 2020) and pandemic (March 2020 to October 2020). Births from November 2020 onwards will be excluded from our regression analysis since the majority were conceived during lockdown, and the pandemic and its socioeconomic consequences might have affected fertility and thereby the characteristics of new families in ways that we cannot fully account for.[20 21 23] In this first approach, we will ignore variation in exposure to mitigation measures during pregnancy and at birth as we aim to estimate the average population-level impact of COVID-19 mitigation measures on pregnancy and birth outcomes.

In our second analytical approach, we will investigate the relationship between the outcomes and exposure to mitigation measures in more detail. As the intensity, duration and timing of exposure to COVID-19 mitigation measures is dependent on the date of conception and duration of pregnancy, each mother and child pair will be given an individually calculated level of cumulative exposure to mitigation measures in Scotland using the SI created by the Oxford COVID-19 Government Response Tracker (OxCGRT).[24] This allows us to estimate a potential dose–response relationship between exposure to mitigation measures as well as potential effect moderation by timing of exposure (focusing on trimesters).

### Databases

We will use linked data from the data sets below:

*National Records of Scotland (NRS) Births*: The NRS holds information on all births registered in Scotland since 1975. These records include information on date and location of the birth and details of the registered parent(s), including their marital/relationship status and their occupational status.

*Scottish Morbidity Record 02 (SMR02)*: SMR02 records all maternity and infant inpatient and day case episodes in Scotland. Around 50% of episodes relate to births and it was these records that were requested for the purposes of the cohort. These include demographic characteristics and information relating to the birth and clinical management.

*NRS and the Scottish Stillbirths and Infant Deaths Survey*: Register of all births, stillbirths and infant (including neonatal) deaths.

*Scottish Birth Records*: All records of a baby's neonatal care in Scotland.

*Community Health Index database*: This contains a unique identifier for all NHS users in Scotland (~99% of population) and is used to link the above data sets.

## Outcomes

We chose outcomes that could feasibly be affected by social mitigation measures (figure 1, logic model) and for their relevance for subsequent child and adult health. We grouped them into maternal behaviours, birth and pregnancy outcomes and service use.

### Maternal behaviours

Smoking in pregnancy, usually measured during the antenatal care booking (~8–12 weeks of pregnancy) supplemented by information collected at any subsequent antenatal appointments (yes; no). Infant feeding at discharge from hospital (breast feeding—yes; no).

### Birth and pregnancy characteristics

Birth weight in grams (continuous variable); low birth weight (LBW) <2500 g and high birth weight (HBW) >4000 g. Similarly, gestational age will be considered as a continuous variable and categorised to identify preterm birth (delivery before $37^{+0}$ weeks of gestation) and late gestational age ($\geq 42^{+0}$ weeks). We will carry out sensitivity analyses differentiating different degrees of prematurity (extremely preterm: $<28^{+0}$ weeks; very preterm: $28^{+0}$–$31^{+6}$ weeks; moderate to late preterm: $32^{+0}$–$36^{+6}$ weeks) and LBW (extremely low: <1000 g; very low: 1000–1499 g; low: 1500–2499 g), since previous research has found delays in extreme prematurity which only manifest in reductions in 'very premature'.[25] Additionally, we will analyse birth weight standardised for gestational age and consequently small for gestational age (SGA) as well as large for gestational age (LGA) as outcomes to explicitly focus on fetal growth. The Apgar score, measured within the first 5 min after delivery, assesses five characteristics (heart rate, respiratory effort, muscle tone, reflex irritability, colour), and can be dichotomised to measure good to excellent infant health (score of 7 or higher[26]). Additionally,

we will examine hypertensive disease of pregnancy by combining International Classification of Diseases 10th Revision (ICD-10) codes for gestational hypertension and pre-eclampsia. We will not examine these outcomes separately as they are clinically closely linked and allocation to ICD-10 codes may vary in precision across areas. Lastly, we will explore pandemic-induced changes in the prevalence of gestational diabetes. However, this outcome is likely affected via changes in the uptake of screening and testing for gestational diabetes during the COVID-19 pandemic.

### Health service use

Mode of delivery will consist of four categories (spontaneous vaginal, assisted vaginal, planned caesarean, emergency caesarean), mode of anaesthesia (spinal, general anaesthesia, epidural) and neonatal unit admissions.

Most of the outcomes under examination are relatively common (eg, rate of preterm births is 65 per 1000). The least common are stillbirths (5 per 1000) and LBW (20 per 1000). With 27 100 births that occurred during the pandemic period (April 2020 to October 2020[27]), these outcomes are relatively infrequent.

### Secondary outcomes

We will also consider changes before/during the pandemic in the following secondary outcomes: miscarriage (loss of baby during first 23 weeks of pregnancy), stillbirths (loss of baby after 24 weeks of gestation) and neonatal deaths (first 28 days after delivery). Some of these outcomes are very rare (eg, neonatal death is <0.2%) and so may only be used to identify bias, with outcome data not reported. Analysis of changes in our secondary outcomes will inform our analysis of postnatal outcomes. If, for example, rates of stillbirths and miscarriages were higher during the pandemic compared with prepandemic periods, we expect the pandemic to have an indirect protective effect on postnatal outcomes via this selection mechanism.

### Exposure

For our first approach—the ITS analysis—we will use dummy variables to indicate whether the outcome (measured at booking or at birth, depending on the outcome) was observed during prepandemic (before first lockdown measures in March 2020) or pandemic periods (April 2020 to October 2020).

For our second analysis, we will calculate an individual level of cumulative exposure for each mother–child pair using the OxCGRT. The OxCGRT has recorded government responses to the COVID-19 pandemic. Methodological details of the OxCGRT have been described elsewhere.[24] As a measure of the stringency of lockdown measures, we will use the OxCGRT SI which comprises nine different indicators (school closing, workplace closing, cancellation of public events, restriction on gathering size, closed public transport, stay-at-home order requirements, restrictions on internal

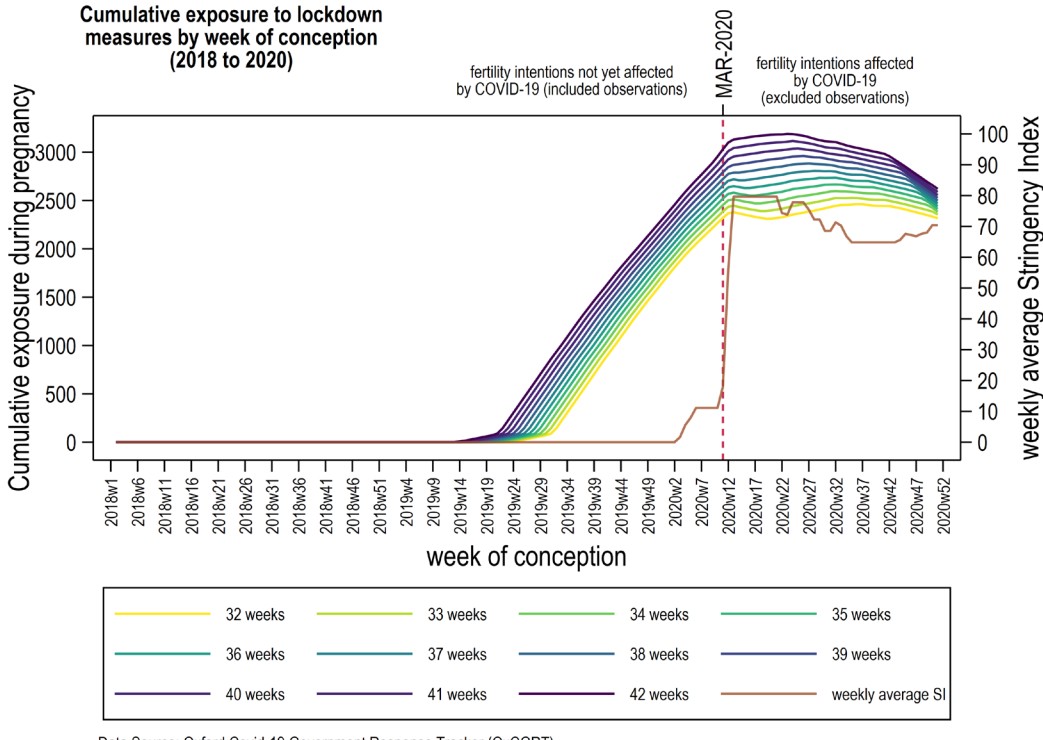

**Figure 2** Visual description of our exposure variable. Sum of weekly average Stringency Index (SI; left vertical axis) during pregnancy for each week of conception (42 weeks of gestation being the top line and 32 weeks of gestation being the bottom line) between January 2018 and December 2020. Level of cumulative exposure is shown for gestational age (32–42 weeks). Crude weekly average SI for Scotland is shown in brown (right vertical axis). Conceptions after March 2020 (indicated by the dashed red line) are excluded from our analyses.

movement, restriction on international travel, public health campaigns). The SI ranges from 0 to 100 and has been recorded daily since January 2020. For Scotland, the SI increased drastically in the first week of March 2020 (SI=11.11) to the highest value during our observation period in the last week of March 2020 (SI=79.63). The time series of weekly average SI is shown in figure 2 (right y-axis). The COVID-19 strategy of the Scottish government can be found at https://www.gov.scot/collections/coronavirus-covid-19-strategic-approach/.

Cumulative exposure to lockdown measures will be calculated by the sum of weekly averages of SI during pregnancy and up until the occurrence of the outcome. Figure 2 visualises the level of cumulative exposure for mother–child pairs by week of conception for different gestational ages. As raised in the Introduction section, it is possible that timing of exposure to social mitigation matters. We will therefore also examine cumulative exposure within each trimester of pregnancy.

### Population characteristics and confounding factors

All models will include dummy variables indicating which month the outcomes were observed (with January being the reference) to account for seasonality and the correlation between month of birth and cumulative exposure.

In the second analytical approach, an association between our cumulative exposure variable and duration of pregnancy arises automatically as mothers with the same conception date but different pregnancy durations will have been exposed to different levels of cumulative exposure at delivery. Therefore, duration of pregnancy will be correlated with the cumulative exposure to SI of a mother–child pair and a postnatal outcome (eg, birth weight) of interest and thus needs to be adjusted for.

Yet, duration of pregnancy is a confounder of the exposure–outcome relationship for postnatal outcomes (because it has a deterministic relationship with our cumulative exposure) and may be a mediator. Exposure to the pandemic might affect gestational age (eg, by changing maternal behaviour or health services) which in turn affects postnatal outcomes (birth weight, Apgar score, neonatal death, infant feeding on discharge, mode of delivery, mode of anaesthesia, neonatal unit admission). Through adjusting for gestational age, we will therefore remove confounding effects but potentially block part of the effect of interest if it is also a mediator. Analyses on gestational age as an outcome will inform the extent of this potential overadjustment for postnatal outcomes.

Change in incidence of miscarriage, pregnancy terminations, stillbirths and maternal emigration behaviour

during pregnancy due to COVID-19 mitigation measures may also act as potential mediators of the exposure–outcome relationship. Because the pandemic might have increased the likelihood of these events, this pathway could potentially result in a protective effect of the exposure on postnatal outcomes (eg, birth weight). Blocking these mediating pathways from exposure to outcome will avoid potentially counteracting, more proximate causes of the association between SI and postnatal outcomes that might deceptively lead to attenuated effects ('live birth bias'). This will be partially achieved by the control variables introduced in model (3), as we expect these characteristics of mother–child pairs (maternal age, sex of baby, maternal National Statistics Socioeconomic Classification (NS-SEC), Scottish Index of Multiple Deprivation (SIMD) and urban-rural classification of residence) to be associated with a potential change in likelihood of these events due to the pandemic. Thus, the resulting estimand is the average total effect of our exposure on postnatal outcomes controlled for potential in utero selection effects. It is not an aim of the study to examine other mediating mechanisms.

We will also adjust for variables that are associated with the outcome but not with the exposure—to take account of potential time trends in outcomes, including, where sufficiently complete, maternal age, maternal occupational class measured by NS-SEC, ethnicity of mother, sex of the baby, SIMD and urban-rural classification of residence. Informed by previous work,[28] we expect a large proportion of missing information on maternal ethnicity (around 50%) but high completeness (>90%) in the other variables.

## Impacts on inequalities

In both approaches, several axes of inequality will be examined to consider whether the impacts of the pandemic have been differential: ethnicity/mother's country of birth (depending on completeness and available sample size), area deprivation (SIMD), urban-rural classification of residence, first-time mothers, maternal social position (NS-SEC) and relationship status of parents (sole registrations, separated, cohabitating, married). We will measure both absolute and relative inequalities.

Relationship status, SIMD and urban-rural classification of residence can possibly change due to COVID-19 mitigation measures. Using our first analytical approach, we will assess potential step or slope changes in the number of births born to mothers in different relationship, SIMD and urban/rural categories following March 2020. As we expect no compositional changes due to selection into pregnancy within our chosen observation period, this analysis will inform to which extent compositional changes regarding area-level characteristics (SIMD and urban/rural classification) were due to maternal moving behaviour.

## Statistical analysis

In the first approach, we will use ITS regression models to describe time trends in the outcomes. Therefore, we will constrain this analysis to linear functions of time. Covariates in these models will be time (weeks or months) since first date of collected data, a dummy variable indicating whether an observation belongs to the exposed or unexposed group, an interaction between time and the exposure dummy variable, and dummy variables indicating in which month the outcome was observed with January being the reference month. Our data are structured by two levels: mother–child pairs nested within small geographic areas. Therefore, we will use multilevel modelling throughout our regression analysis. Model (1) exemplarily shows the formal specification for the continuous outcome birth weight $y_{ij}$ of mother–child pair $i$ nested within small area (data zone) $j$. For non-continuous outcomes (smoking, infant feeding, LBW, HBW, prematurity, SGA, LGA, method of delivery, mode of anaesthesia, pre-eclampsia, neonatal admissions, stillbirth, neonatal death), we will use weekly prevalence rate (number of weekly events/number of weekly live births). For the least common outcomes (stillbirth and LBW), we will use monthly prevalence rates if necessary.

$$y_{ij} = \beta_0 + \beta_1 week_{ij} + \beta_2 exposed_{ij} + \beta_3 week_{ij} X exposed_{ij}$$
$$+ \sum_{t=1}^{11} \beta_t month_{ij} + u_{0j} + \varepsilon_{0ij}, \ u_{0j} \sim N\left(0, \sigma_{u0}^2\right), \ \varepsilon_{0ij} \sim N\left(0, \sigma_{e0}^2\right) \quad (1)$$

In the second approach, the exposure is the cumulative SI and we will adjust for potential confounders. As an example, we formally describe our models for the continuous outcome birth weight below.

$$y_{ij} = \beta_0 + \beta_1 SI_{ij} + \beta_2 DoP_{ij} + \sum_{t=1}^{11} \beta_t month_{ij}$$
$$+ u_{0j} + \varepsilon_{0ij}, \ u_{0j} \sim N\left(0, \sigma_{u0}^2\right), \ varepsilon_{0ij} \sim N\left(0, \sigma_{e0}^2\right) \quad (2)$$

Model (2) presents our most parsimonious model specification, where $y_{ij}$ is birth weight (in grams) measured for mother–child pair $i$ in data zone $j$, $SI_{ij}$ is the sum of weekly average Stringency Index during pregnancy of mother–child pair $ij$, $DoP_{ij}$ is the duration of pregnancy (in weeks) for mother–child pair $ij$, and $month_{ij}$ is a dummy variable that indicates in which month birth was given with January being the reference category. In model (3), we further include the neutral control variables maternal age, sex of baby, maternal NS-SEC, SIMD and urban-rural classification of residence. In case there is considerable missing information in a neutral control variable, we will omit it from our models as the risk of bias induced by missing not at random likely outweighs the potential gains of a neutral control.

$$y_{ij} = \beta_0 + \beta_1 SI_{ij} + \beta_2 DoP_{ij} + \sum_{t=1}^{11} \beta_t month_{ij}$$
$$+ \beta_3 age_{ij} + \beta_4 sex_{ij} + \beta_5 NSSEC_{ij} + \beta_6 SIMD_{ij} \quad (3)$$
$$+ \beta_7 urban_{ij} + u_{0j} + \varepsilon_{0ij}, \ u_{0j} \sim N\left(0, \sigma_{u0}^2\right), \ \varepsilon_{0ij} \sim N\left(0, \sigma_{e0}^2\right)$$

Moreover, for postnatal outcomes, we will explore whether timing of exposure matters by including variables for cumulative exposure during each trimester of

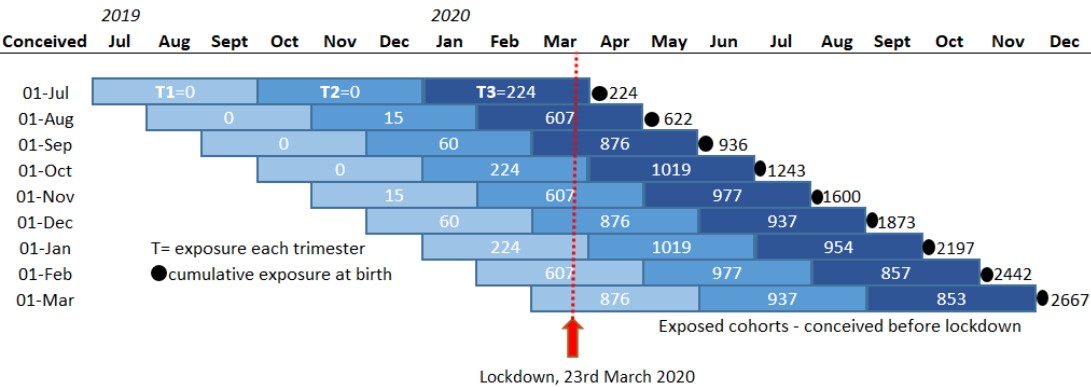

**Figure 3** Exposed groups under investigation. Cumulative levels of exposure presented here are the sum of weekly averages of the Stringency Index within each trimester up to month of birth. In this figure, conceptions and births are assumed to occur on the first of each month with equal gestational age. Note that, in the analyses, cumulative exposure is calculated for each mother–child pair individually and, thus, these exposure levels do not match those in figure 2.

pregnancy as shown in model (4), where $SI1_{ij}$, $SI2_{ij}$, $SI3_{ij}$ is the sum of weekly average Stringency Index during the first, second and third trimesters of pregnancy of mother–child pair $ij$. Figure 3 illustrates the cumulative exposure to mitigation measures during each trimester. As our data do not include cohorts that experienced high levels of exposure during their first trimester and low exposure during their third trimester, we will only test differences in the effect of exposure during the third and second trimesters.[22]

$$
\begin{aligned}
y_{ij} = {} & \beta_0 + \beta_1 SI1_{ij} + \beta_2 SI2_{ij} + \beta_3 SI3_{ij} + \beta_4 DoP_{ij} \\
& + \sum_{t=1}^{11} \beta_t month_{ij} + \beta_5 age_{ij} + \beta_6 sex_{ij} + \beta_7 NSSEC_{ij} + \beta_8 SIMD_{ij} \quad (4) \\
& + \beta_9 urban_{ij} + u_{0j} + \varepsilon_{0ij},\ u_{0j} \sim N\left(0, \sigma_{u0}^2\right),\ \varepsilon_{0ij} \sim N\left(0, \sigma_{e0}^2\right)
\end{aligned}
$$

To help with the interpretation of our results, we will present the estimated values for hypothetical plausible levels of cumulative exposure (ie, exposure to 3, 6, 9 months of an SI of 50, 60, 70, etc). In addition, we will present the estimated average outcome values at specific time points. Note that no mother–child pair in our data has experienced a different level of exposure at the same date of birth and length of gestation.

We will examine differential effects by including interaction terms with the modifying variables (see Impact on inequalities section). Where interactions appear to be meaningful, we will stratify the models. Inequalities in effect sizes will be examined by comparing the average effects between levels of moderating variables. In model (5), we exemplarily show the specification of such a model for inequalities in the effect of our exposure variable along parental NS-SEC.

$$
\begin{aligned}
y_{ij} = {} & \beta_0 + \beta_1 SI_{ij} + \beta_2 DoP_{ij} + \sum_{t=1}^{11} \beta_t month_{ij} + \beta_3 age_{ij} \\
& + \beta_4 sex_{ij} + \beta_5 NSSEC_{ij} + \beta_6 SIMD_{ij} + \beta_7 urban_{ij} + \beta_8\ SI_{ij} \quad (5) \\
& x\ NSSEC_{ij} + u_{0j} + \varepsilon_{0ij},\ u_{0j} \sim N\left(0, \sigma_{u0}^2\right),\ \varepsilon_{0ij} \sim N\left(0, \sigma_{e0}^2\right)
\end{aligned}
$$

Multilevel linear models will be used for continuous outcomes (as appropriate for the distribution of outcome data), with multilevel binary and multinomial logistic regression models used for binary and categorical outcomes, respectively. All models will be estimated by maximum likelihood. We will derive prevalence ratios and absolute differences from model estimates.

### Sensitivity analysis

In our ITS regression analysis, the included linear time trend and month indicator variables may not fully address the autocorrelation of observations. We will therefore inspect the autocorrelation function and partial autocorrelation function of our model residuals and resort to (seasonal) autoregressive integrated moving average models if necessary.

We will explore non-linearity in the effect of cumulative exposure to lockdown measures by re-estimating our models with a quadratic functional form of the exposure variable as well as a semiparametric specification, in which we use quintiles of the exposure variable as cut-offs to form discrete levels of cumulative exposure. We will repeat analyses limited to singleton births. Additionally, we will analyse induced and spontaneous preterm births separately (if sample size is sufficient). Depending on the partnership status of parents at birth registration, we will also have information on paternal NS-SEC. We will conduct sensitivity analyses in which we exchange maternal with paternal NS-SEC where available, as well as taking the higher occupational class in the household.

As noted previously, excluding births conceived during lockdowns will reduce unmeasured or residual confounding due to changed sociodemographic parental characteristics likely associated with the outcomes.[20 21] However, changes in the likelihood of miscarriage, pregnancy terminations, stillbirths, neonatal deaths and maternal emigration behaviour during pregnancy may still introduce bias for postnatal outcomes. We will explore this by analysing time trends for available variables (stillbirth, miscarriage, neonatal death) using ITS regression as described above. If this

analysis suggests that our exposure–outcome relationship is susceptible to such potential selection bias, we will further control (where possible) for variables that are likely associated with miscarriage, pregnancy terminations, stillbirths and maternal emigration behaviour during pregnancy as well as the outcomes (but not affected by the exposure).

Finally, we will explore unmeasured confounding by splitting our data in multiple unexposed comparison groups (April to October for each year between 2010 and 2019).[29] Systematic differences in our outcomes between unexposed groups conditional on the covariates listed above will be tested by estimating the effect of dummy variables indicative of which comparison group a mother–child pair belongs to using regression analyses. Systematic differences in the outcomes between unexposed comparison groups even after adjusting for our set of covariates will reveal whether there is potential unmeasured confounding in respect to the effect of our cumulative exposure variable on the outcomes. The exposure–outcome relationship will then be estimated using varying sets of unexposed comparison groups against the exposed group of mother–child pairs (April to October 2020). Resulting effect sizes will be shown in forest plots and a pooled effect will be estimated by random-effects meta-analysis. In case unexposed comparison groups indeed differ in respect to our outcomes after covariate adjustment, results of this pooled analysis will be interpreted in light of unexplainable differences between unexposed groups.

### Sample size
Our sample consists of all child and mother pairs for children born in Scotland between March 2010 and October 2020. Sample size is expected to be n~500 000 mother–child pairs (estimated based on an average of 50 000 births per annum).

### Missing data
We will document levels of missing data in all variables of interest, over time and according to the potential effect moderators, for two reasons. First, understanding how data collection was impacted during the early stages of the pandemic can inform responses to future pandemics. Second, changes in patterns of missingness in the data, due to the pandemic, could introduce bias. In case of considerable levels of missing data, item missingness will be addressed using multiple imputation by chained equations.

### Ethics and dissemination
Use of the data has been approved by the Public Benefit and Privacy Panel for Health and Social Care. Results of this research will be disseminated in peer-reviewed presentations at public health national and international conferences and open-access, peer-reviewed journal articles. We will produce a briefing paper for policy makers and practitioners and will work with in-house press advisors to ensure visibility in newspapers, radio, etc and on our COVID-19 Unit web page.

**Author affiliations**
[1]MRC/CSO Social and Public Health Sciences Unit, School of Health and Wellbeing, University of Glasgow, Glasgow, UK
[2]Department of Social and Preventive Medicine, Center for Public Health, Medical University of Vienna, Wien, Austria
[3]Public Health Scotland, Glasgow, UK
[4]Health Data Science Research Centre, University of Aberdeen, Aberdeen, UK
[5]Usher Institute, University of Edinburgh, Edinburgh, UK
[6]Dentristy and Nursing, School of Medicine, University of Glasgow, Glasgow, UK

**Contributors** RD, AHL, MO and AP conceived the study. SMN obtained the data and associated approvals. PMH, SP, SJS, RW, SMN and RK contributed to the conception of the study design. MO, AP and PMH drafted the study protocol. RD, AHL, PMH, SP, SJS, RW, SMN and RK provided critical feedback on the draft manuscript and approved the final version.

**Funding** The work of the authors was supported by the Medical Research Council (MC_UU_00022/2) and the Scottish Government Chief Scientist Office (SPHSU17). AP also receives support from the Wellcome Trust (205412/Z/16/Z). MO was also supported by the Marietta Blau Scholarship (MPC-2021-00178; funded by the Austrian Federal Ministry of Education, Science and Research). AHL, AP and RD are members of, and receive support from, the UK Prevention Research Partnership Maternal and Child Health Network (MR/S037608/1).

**Competing interests** None declared.

**Patient and public involvement** Patients and/or the public were not involved in the design, or conduct, or reporting, or dissemination plans of this research.

**Patient consent for publication** Not applicable.

**Provenance and peer review** Not commissioned; externally peer reviewed.

**ORCID iDs**
Moritz Oberndorfer http://orcid.org/0000-0003-3987-0823
Shantini Paranjothy http://orcid.org/0000-0002-0528-3121
Sarah Jane Stock http://orcid.org/0000-0003-4308-856X
Rachael Wood http://orcid.org/0000-0003-4453-623X

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
