## [Reviewer comments · BMJ Open]

ARTICLE DETAILS

TITLE (PROVISIONAL)	Study Protocol: Examining the impacts of COVID-19 mitigation measures on pregnancy and birth outcomes in Scotland: A linked administrative data study
AUTHORS	Oberndorfer, Moritz; Henery, Paul; Dundas, Ruth; Leyland, Alastair; Paranjothy, Shantini; Stock, Sarah; Wood, Rachael; Nelson, Scott; Kearns, Rachel; Pearce, Anna

VERSION 1 – REVIEW

REVIEWER	Homer, Caroline University of Technology Sydney, Centre for Midwifery, Child and Family Health
REVIEW RETURNED	12-Sep-2022

GENERAL COMMENTS	Thank you for the opportunity to review this protocol. The project uses population based data to track the indirect impacts of the COVID-19 mitigation strategies on maternal and newborn health outcomes. The approach and analysis plan seems sound. The team have expertise in this area and have access to the data. The study will be interesting as it can look at different levels of lockdown and the impacts. I would like to see a clear aim. The aim at the moment is very broad and then 2 analytical approaches are outlined. I suggest having clear aim and objectives in the Abstract and in the main paper. I am interested in how the longitudinal impacts will be assessed. For example, if the stringency index was very high in mid pregnancy but the baby not born until term – how will the impact be assessed? The dose response – the longer the exposure in the lockdowns – is also being considered but given pregnancies are 9 months – how will this issue be addressed especially if the birth takes place sometime after the stringency period?
--

REVIEWER	Eltonsy, Sherif University of Manitoba, College of Pharmacy
REVIEW RETURNED	01-Nov-2022

GENERAL COMMENTS	Please find attached.
-----------------------

VERSION 1 – AUTHOR RESPONSE

Reviewer 1:

Comment 1:

Thank you for the opportunity to review this protocol.

The project uses population-based data to track the indirect impacts of the COVID-19 mitigation strategies on maternal and newborn health outcomes. The approach and analysis plan seems sound. The team have expertise in this area and have access to the data. The study will be interesting as it can look at different levels of lockdown and the impacts.

Response 1:

We thank the reviewer for complimenting our analysis plan and the expected contribution of this study.

Comment 2:

I would like to see a clear aim. The aim at the moment is very broad and then 2 analytical approaches are outlined. I suggest having clear aim and objectives in the Abstract and in the main paper.

Response 2:

We have stated the aim more explicitly and have amended the introduction accordingly:

Introduction p.5:

“We aim to estimate the wider impacts of the COVID-19 pandemic on pregnancy and birth outcomes and inequalities in Scotland. More specifically, we aim to estimate changes in health and pregnancy outcomes as a result of the pandemic.”

...

“As part of this aim, we will investigate whether exposure to mitigation measures had a differential effect on our outcomes across several axes of inequalities. Second, we aim to consider the cumulative effects of social mitigation measures across pregnancy. We will use the stringency index (which measures the strictness of policies that primarily restrict people’s behaviour) and compare cohorts with different lengths or intensity of exposure. Additionally, we aim to consider timing of exposure, as it is possible that, for some outcomes, any impacts of the stresses related to the pandemic and social mitigation measures might be greater during some trimesters of pregnancy than others²¹.”

Comment 3:

I am interested in how the longitudinal impacts will be assessed. For example, if the stringency index was very high in mid pregnancy but the baby not born until term – how will the impact be assessed?

Response 3:

In Model (4), we will assess if the same level of exposure to stringency index has a different effect on the outcomes depending on the timing (trimester) of exposure during pregnancy using the model specification below for (exemplarily) birth weight.

(4)

In Figure 3, we have visualised how we separate the cumulative exposures into trimesters: Weekly averages of the Stringency Index (which is recorded for every day) are summed starting from conception to end of the first trimester, over the second trimester, and over the third trimester. However, the data only allows us to estimate differences in the effect of SI during second and third trimester (statistical analysis section). Figure 2 also shows the weekly average SI (right y-axis) indicating that the Stringency Index remains on a high level (60 to 80) for the entire cohort of exposed mother-child pairs in our sample. Therefore we cannot estimate if the effect of exposure

during the first trimester was different compared with exposure in the second or third trimester (statistical analysis section).

Comment 4:

The dose response – the longer the exposure in the lockdowns – is also being considered but given pregnancies are 9 months – how will this issue be addressed especially if the birth takes place sometime after the stringency period?

Response 4:

As visualised in figure 2, our measure of cumulative exposure is driven by two variables. First, the sum of weekly average Stringency Index (SI) since the week of conception. The closer conception was to the first introduction of COVID-19 mitigation measures in March 2020, the longer a mother-child pair was exposed to mitigation measures during pregnancy. Second, as the cumulative exposure is a sum of SI over the entire pregnancy period, longer pregnancies will have higher cumulative exposures even if date of conception was identical. Therefore, cumulative exposure will be highest for mother-child pairs who conceived just before March 2020 and had the longest duration of pregnancy (figure 2). In our models, we include duration of pregnancy to adjust for this confounding factor.

Reviewer 2:

Comment 5:

Title: It may be better to specify that the study is conducted in Scotland or using Scotland linked administrative data

Response 5:

Thank you for this suggestion. The title now reads:

“Study Protocol: Examining the impacts of COVID-19 mitigation measures on pregnancy and birth outcomes in Scotland: A linked administrative data study”

Comment 6:

Abstract: (Page 2, lines 33-36) *“estimating a potential dose-response relationship between exposure to mitigation measures and our outcomes of interest as well as potential effect moderation by timing of exposure during pregnancy.”* However, in the full text, the authors did not explain about effect modification analysis. Please clarify the analysis method for effect modification if it is used or remove from abstract.

Response 6:

We explain our approach to assessing if the timing of exposure matters for the outcome when describing Model (4) and illustrate the separation of our exposure variable in figure 3. The reviewer is right in saying that we do not explicitly refer to this analysis as moderation analysis in the full text. We thus changed our wording in the abstract:

“Thus, estimating a potential dose-response relationship between exposure to mitigation measures and our outcomes of interest as well as assessing if timing of exposure during pregnancy matters.”

Comment 7:

Introduction: *The authors mentioned “In Canada, new-born readmission rates among first time mothers were higher after the pandemic, while multiparous women were less likely to experience pre-term birth rates, low Apgar scores and hospital readmissions.19” (page 5, lines 28-32). However, this referenced study (ref. 19) is not conducted in Canada, it’s conducted in USA. Please check, clarify, and confirm the included reference studies.*

Response 7:

Thank you for catching this. The correct study has been cited but we somehow mentioned the wrong country.

“In the United States, new-born readmission rates among first time mothers were higher after the pandemic, while multiparous women were less likely to experience pre-term birth rates, low Apgar scores and hospital readmissions.19”

Comment 8:

Introduction: The authors mentioned “We will take a natural experiment approach to identify any step change trends in outcomes at the start of the pandemic, limiting our analyses to pregnancies which were conceived before the pandemic, to avoid introducing bias due to the changing socio-demographic characteristics of conceptions which occurred after the start of the pandemic 20.” (Page 5, lines 39-44). **And in methods** (page 6, lines 21- 23) “Births from November 2020 onwards will be excluded from our regression analysis since the majority were conceived during lockdown, and the pandemic”. However, study reference 20 stated that “We can indeed observe a sudden change in the proportion of births in the most and least deprived areas of Scotland after November/ December 2020.” Please clarify and state the reasons of why you excluded pregnancies starting in November while evidence state that observed changes in fertility were seen after November/ December 2020. **Also, in methods Exposure section** (page 8, lines 34-37) the authors mentioned “We will additionally include November and December 2020 in our visualisations (if this data becomes available at time of analysis) but restrict our modelling to observations up until October 2020.” Why the authors decided to include the births (visualization) during November and December 2020? Authors should be clear on the included study period as readers may be puzzled.

Response 8:

Thank you for your careful considerations about the study population. We expect compositional changes to reach their largest extent following November 2020 as these births will have been conceived during lockdown. However, it is possible that we may see compositional changes in parental characteristics for births occurring in November (and possibly even October) as parents may have changed their family planning in light of knowledge of the COVID-19 outbreak in China. By excluding births from November onwards (and potentially October as may be advised by the results of our analysis of pre-natal outcomes) we also make sure that compositional changes due to changes in abortions cannot strongly confound our associations.

We agree that including births in November and December in visualisations may be confusing and should be subject to further, separate research. We have thus removed this part of the study protocol.

Comment 9:

Methods, study design and population: The authors did not mention the design nor clear definitions of the included population (page 6). The authors should explicitly define the included pregnancy cohort that will be used.

Response 9:

We now make the population of interest more explicit at the beginning of this section:

“Our study population includes live births born between March 2010 and October 2020. More precisely, our population of interest consists of live births conceived before the pandemic who have not been exposed to COVID-19 mitigation measures in utero (live births between March 2010 and February 2020) and those who were conceived before the pandemic but were exposed to mitigation measures in utero (live births between March and October 2020). “

Comment 10:

Methods, population characteristics and confounding factors section: The authors mentioned “will adjust for variables that are associated with the outcome but not with the exposure...” (page 9, lines 24-32). However, they did not mention if they will examine the maternal chronic conditions as hypertension, diabetes, asthma, etc. Since these conditions are considered risk factors for negative pregnancy outcomes as preterm birth or stillbirth. Will the authors consider previous preterm delivery, stillbirth or caesarean delivery as confounding factor? Please clarify why did not include these conditions.

Response 10:

Thank you for these suggestions. Long-term conditions of mothers may act as moderators of the pandemic on pregnancy and birth outcomes. However, we do not believe them to be confounders as they are unlikely to have affected the exposure and indeed are possibly more likely to be affected by the exposure, making them mediators. For the Lockdown Cohort (i.e. who were conceived during lockdown), however, maternal chronic conditions and hypertension could have been involved in the selection into pregnancy during the COVID-19 pandemic. To avoid these potential

selection biases, we have excluded babies conceived during the pandemic. If, however, the pandemic has increased the rate of miscarriages/abortions/stillbirths among mothers with chronic conditions but not among mothers without these conditions, our estimates may be biased. We discuss this 'live birth bias' in the section "Population characteristics and confounding factors".

Comment 11:

Methods, outcomes: *It is important to provide clear definitions of birth and pregnancy outcomes. The authors should state the definition reference for these outcomes. Also, why did the authors only include preeclampsia as an outcome? What about other conditions like gestational diabetes or gestational hypertension? Please clarify.*

Response 11:

Thank you for this important question.

We have discussed the anticipated completeness and recording of gestational diabetes and gestational hypertension with our clinical collaborators. We have decided to combine pre-eclampsia and gestational hypertension as they are clinically closely linked and allocation to respective ICD10 codes may vary between areas. The combined outcome is hypertensive disease of pregnancy. Gestational diabetes is also available through the Scottish Morbidity Record 02 and thus could be included in our analyses. However, there is non-systematic screening of gestational diabetes, and we thus cannot distinguish between the pandemic's effect on the prevalence of gestational diabetes from the pandemic's effect on screening or testing of gestational diabetes through antenatal care. We have expanded our outcomes section on page 8 accordingly.

"Additionally, we will examine hypertensive disease of pregnancy by combining ICD10 codes for gestational hypertension and pre-eclampsia. We will not examine these outcomes separately as they are clinically closely linked and allocation to ICD10 codes may vary in precision across areas. Lastly, we will explore pandemic-induced changes in the prevalence of gestational diabetes. However, this outcome is likely affected via changes in the uptake of screening and testing for gestational diabetes during the COVID-19 pandemic."

Comment 12:

Methods, outcomes: *(page 7, lines 55-57) Health services use: Mode of delivery will consist of four categories (spontaneous vaginal, assisted vaginal, planned caesarean, emergency caesarean), mode of anaesthesia (spinal, general anaesthesia, epidural). How will the authors differentiate between these outcomes: spinal, general anaesthesia, epidural and spontaneous vaginal, assisted vaginal. How the authors think that pandemic restrictions impacted the choice between assisted and spontaneous vaginal delivery or between epidural and spinal anaesthesia? The authors should reconsider to include vaginal (assisted or spontaneous) versus the caesarean section and anaesthesia (spinal/epidural vs. general).*

Response 12:

Thank you for these questions. Our main hypothesis was that method of delivery may have been affected as the COVID-19 pandemic changed the availability of hospital resources and thus potentially led to changed hospital practice. However, it is also possible that changes to pregnancy characteristics (e.g. gestational age) may have also impacted upon on mode of delivery. Both may have changed timings of assisted vaginal deliveries for example. Since we are still learning about how things changed during the pandemic, our preference would be to not collapse variables unless sample sizes require this. Unfortunately, we cannot know whether this is necessary until we have seen the data.

Comment 13:

Methods, Exposure: *This section does not describe the different mitigation measures implemented in Scotland. Briefly explain the implemented restrictions specific for Scotland by calendar time to clarify the exact intervention date.*

Response 13:

Thank you for pointing this out.

We now refer the reader more explicitly to our figure to wherein we illustrate our cumulative exposure variable as well as the time series of the Stringency Index for Scotland (right y-axis in figure 2). We

additionally mention dates and provide a link to the Scottish government's official COVID-19 strategy for a more detailed explanation of Scottish mitigation measures. The SI is briefly explained in the method section with a reference pointing the reader to a more detailed explaining of the SI.

"For Scotland, the SI increased drastically between the first week of March 2020 (SI=11.11) to the highest value during our observation period in the last week of March 2020 (SI=79.63). The time series of weekly average SI is shown in figure 2 (right y-axis). The detailed COVID-19 strategy of the Scottish government can be found at <https://www.gov.scot/collections/coronavirus-covid-19-strategic-approach/>."

Comment 14:

Methods, population characteristics and confounding factors section: *In the second analytic approach, duration of pregnancy will be associated with cumulative exposure to mitigations measures and the outcomes and will thus be adjusted for. Yet, duration of pregnancy may not only be a confounder of the exposure-outcome relationship for postnatal outcomes, but also a mediator (page 9, lines 11-18). The authors should explicitly explain how the duration of pregnancy is a mediator in this case, especially that the outcome examined is weeks of gestation. Also, the authors mentioned twice (page 9, lines 16 and 39) about mediation analyses although not conducting this analysis, so readers may be puzzled when interpreting these sentences. It may be better to specify the exact analysis plan in the method section and to add the mediation analysis as a limitation.*

Response 14:

We appreciate this comment and try to clarify.

We repeatedly mention duration of pregnancy in its role as both a confounder and a mediator of the exposure –post-natal outcomes relationship in the confounding section and sensitivity analysis section of the protocol. In line with the reviewer's suggestions, we have altered the "*Population characteristics and confounding factors*" p.9. We are now more explicit about how duration of pregnancy is a confounder and potentially mediator at the same time and how we plan to deal with this challenge:

"In the second analytic approach, an association between our cumulative exposure variable and duration of pregnancy arises automatically as mothers with the same conception date but different pregnancy durations will have been exposed to different levels of cumulative exposure at delivery. Therefore, duration of pregnancy will be correlated with the cumulative exposure to SI of a mother-child pair and a post-natal outcome (e.g., birthweight) of interest and thus is a confounder that needs to be adjusted for.

Yet, duration of pregnancy is not only a confounder of the exposure-outcome relationship for post-natal outcomes (because it has a deterministic relationship with our cumulative exposure) but may also be a mediator. Exposure to the pandemic might affect gestational age (e.g., by changing maternal behaviour or health services) which in turn affects post-natal outcomes (birthweight, Apgar score, neonatal death, infant feeding on discharge, mode of delivery, mode of anaesthesia, neonatal unit admission). Through adjusting for gestational age, we will therefore remove confounding effects but potentially block part of the effect of interest if it is also a mediator. Analyses on gestational age as an outcome will inform the extent of this potential overadjustment for post-natal outcomes.

Change in incidence of miscarriage, pregnancy terminations, stillbirths, and maternal emigration behaviour during pregnancy due to COVID-19 mitigation measures may also act as potential mediators of the exposure-outcome relationship. Because the pandemic might have increased the likelihood of these events, this pathway could potentially result in a protective effect of the exposure on post-natal outcomes (for example birthweight). Blocking these mediating pathways from exposure to outcome will avoid potentially counteracting, more proximate causes of the association between SI and post-natal outcomes that might deceptively lead to attenuated effects ('live birth bias'). This will be partially achieved by the control variables introduced in Model (3), as we expect these characteristics of mother-child pairs (maternal age, sex of baby, maternal NS-SEC, SIMD, and urban-rural classification of residence) to be associated with a potential change in likelihood of these events due to the pandemic. Thus, the resulting estimand is the average total effect of our exposure on post-

natal outcomes controlled for potential in utero selection effects. It is not an aim of the study to examine other mediating mechanisms.”

Comment 15:

Methods, impact on inequalities: *the authors mentioned that first time mothers as one of the inequalities. Please briefly explain how, or reconsider it as a variable and not an inequality factor. Also, relationship status of parents (sole registrations, separated, cohabitating, married) this factor could change during the study period. How did the authors define this variable and will adjust for it?*

Response 15:

We believe the impacts of social mitigation measures may have been greater for first time parents, because reduced contact with social networks and the removal of face-to-face antenatal classes would be potentially more important sources of emotional and practical support among first time parents.

Data on relationship status is collected through the birth certificate which occurs within 21 days of birth. We opt to not adjust for relationship status of mothers, as it is a potential moderator and not a confounder of our exposure-outcome relationship. The reviewer is right in pointing out that exposure to the pandemic may have affected relationship status during pregnancy. If this were the case, relationship status could act as a mediator of the exposure-outcome relationship. Adjusting for this mediator would change our estimand as now better explained in the “*Population characteristics and confounding factors*”.

In light of the reviewer’s comment, we will additionally assess whether there are step or slope changes (see analytical approach 1 in the study protocol) in the number of births born to mothers in different relationship categories following March 2020. Using the same analytical approach, we will also assess whether social mitigation measures were associated with internal migration to more or less deprived areas. As we expect no compositional changes within our chosen observation period, this will inform to which extent compositional change regarding area level characteristics (SIMD and urban/rural classification) were due to moving behaviour.

“Impact on inequalities” p.10

“Relationship status, SIMD, and urban-rural classification of residence can possibly change due to COVID-19 mitigation measures. Using our first analytical approach, we will assess potential step or slope changes in the number of births born to mothers in different relationship, SIMD, and urban/rural categories following March 2020. As we expect no compositional changes due to selection into pregnancy within our chosen observation period, this analysis will inform to which extent compositional change regarding area level characteristics (SIMD and urban/rural classification) were due to maternal moving behaviour.”

Comment 16:

Methods, statistical analysis: *Why did the authors choose to use weekly rates instead of monthly or quarterly? Especially with some outcomes that are infrequent (as stillbirth and LBW) as mentioned by the authors. Please clarify how you will use the weekly rates without affecting the study precision.*

Response 16:

This applies to our interrupted time series regression analysis. Our exemplary Model (1) for this analysis is based on birth weight and thus we have not been clear on how we will treat event data like stillbirths or LBW. We added another paragraph on p.11:

“For non-continuous outcomes (smoking, infant feeding, LBA, HBW, prematurity, SGA, LGA, method of delivery, mode of anaesthesia, preeclampsia, neonatal admissions, stillbirth, neonatal death), we will use weekly prevalence rate (number of weekly events/number of weekly live births). For the least common outcomes (stillbirth and LBW), we will use monthly prevalence rates if necessary.”

Comment 17:

Methods, statistical analysis: *Please clarify if the authors will consider the autocorrelation between the examined points. Will you consider autoregressive terms? Or use autocorrelation plots and other appropriate tests?*

Response 17:

We anticipate addressing autocorrelation by adjusting for monthly seasonality. However, we will additionally check if this is the case by inspecting the autocorrelation function and partial autocorrelation function in line with the reviewer's suggestions. We added the following paragraph to the section on sensitivity analysis p.13:

"In our ITS regression analysis, the included linear time trend and month indicator variables may not fully address the autocorrelation between observations. We will therefore inspect the autocorrelation function and partial autocorrelation function of our time series and resort to (seasonal) Autoregressive Integrated Moving Average (SARIMA) Models if necessary."

Comment 18:

It would be better flow if the authors mention the method section as follows: Patient and public involvement, Databases, Study design and population, population characteristics and confounding, exposure, outcomes, impact on inequalities, statistical analysis, and sensitivity analysis.

Response 18:

The structure of the study protocol follows BMJ Open's guidelines, but we would be happy to restructure as suggested if the editor prefers

Comment 19:

Typing: (Page 6, line 43) focusing rather than focussing on trimesters.

Response 19:

Thank you for catching this. We have corrected the typo.

VERSION 2 – REVIEW

REVIEWER	Eltonsy, Sherif University of Manitoba, College of Pharmacy
REVIEW RETURNED	25-Dec-2022
GENERAL COMMENTS	The authors addressed the comments clearly. No additional comments to consider.